

# Conservation implications of herbicides on seagrasses: sublethal glyphosate exposure decreases fitness in the endangered *Zostera capensis*

Johanna W. van Wyk[1], Janine B. Adams[2] and Sophie von der Heyden[1]

[1] Department of Botany and Zoology, Stellenbosch University, Matieland, South Africa
[2] Institute for Coastal and Marine Research, Department of Botany, Nelson Mandela University, Gqeberha, South Africa

## ABSTRACT

Worldwide seagrass populations are in decline, calling for urgent measures in their conservation. Glyphosate is the most widely used herbicide globally, leading to increasing concern about its ecological impact, yet little is known about the prevalence or impact of glyphosate on seagrasses. In this study, we investigated the effect of sublethal glyphosate exposure on the endangered seagrass, *Zostera capensis*, to identify effects on growth, photosynthetic pigments and leaf morphology as measures of seagrass fitness. Seagrasses were exposed to a single dose of a commercial glyphosate formulation—ranging between 250 to 2,200 µg/L. After three weeks, the median leaf area decreased by up to 27%, with reductions of up to 31% in above ground biomass ($p < 0.05$). Photosynthetic pigment concentration showed no significant difference between groups. The observed effects on biomass and leaf area were seen at glyphosate levels below the regulatory limits set for surface water by several countries and may negatively affect the long-term resilience of this ecosystem engineer to additional stressors, such as those associated with climate change and anthropogenic pollution. As such, glyphosates and other herbicides that are washed into estuarine and marine ecosystems, pose a significant threat to the persistence of seagrasses and are important factors to consider in seagrass conservation, management and restoration efforts.

## INTRODUCTION

Coastal vegetated ecosystems, including seagrass meadows, mangroves, and saltmarshes, play a vital role in sustaining marine biodiversity fisheries (*Perkins-Visser, Wolcott & Wolcott, 1996*; *Verweij et al., 2008*; *Jänes et al., 2020*). In addition, coastal vegetated systems are expected to play an increasingly important role in mitigating the impacts of climate change over the next decades, due their well-documented contributions to coastal protection (*Ondiviela et al., 2014*) and carbon sequestration (*Fourqurean et al., 2012*; *Githaiga et al., 2017*; *Hilmi et al., 2021*). Seagrass meadows, in particular, are known to be

Corresponding author
Sophie von der Heyden,
svdh@sun.ac.za

excellent carbon traps, especially in sediment, sequestering up to 30× more carbon than rainforest areas of similar size, and are estimated to contain between 4.2 and 8.4 PG carbon on a planetary scale (*Fourqurean et al., 2012*). In addition, seagrass meadows reduce the detrimental impact of pollutants and excess nutrients on estuaries and shallow marine environments (*Levine et al., 1990*; *Olisah et al., 2021*). However, seagrass meadows are globally in decline, as demonstrated by *Waycott et al. (2009)* who documented a decline in cover of approximately 7% per year since 1990, leading to urgent calls to increase seagrass conservation and management efforts.

Factors known to contribute to loss in seagrass cover include increased physical disturbance due to boating activity and bioturbation, as well as increased turbidity and eutrophication due to agricultural runoff (*Burkholder, Tomasko & Touchette, 2007*; *Waycott et al., 2009*; *Mvungi & Pillay, 2019*). Although eutrophication and decreased light availability (*Ralph et al., 2007*) due to turbidity are well-described drivers of seagrass decline, this is not the only impact agriculture has on seagrass meadows. Interest in the effects of herbicide and other pollutant runoff on this ecosystem has increased as ever-increasing environmental chemical loads become more evident (*Persson et al., 2022*). At this stage, the effects of pollutants on African seagrass species are poorly represented in the global literature, with research concentrated on Europe, the Americas and Australasia (*Fraser & Kendrick, 2017*; *Gamain et al., 2018*; *Arcuri & Hendlin, 2019*).

The Cape eelgrass, *Zostera capensis*, is the dominant seagrass species in South Africa, occurring in isolated shallow bays and estuaries along the coastline (*Adams, 2016*), with isolated populations also occurring in Mozambique and southern Kenya (*Phair et al., 2019*), but throughout its range it is poorly protected (*van Niekerk et al., 2019*). As of 2021, this species has been classified as endangered due to persistent population declines and the total loss of meadows in certain estuaries (*Adams, 2016*; *van Niekerk et al., 2019*), with associated losses in evolutionary resilience (*Phair et al., 2020*). Although some work has been done investigating regional declines, such as examining the impact of eutrophication on *Zostera capensis* (*Mvungi & Pillay, 2019*), very little is known about other contributing factors.

Even though glyphosate is now the most widely used herbicide globally by volume (*Cuhra, Bøhn & Cuhra, 2016*; *Maggi et al., 2019*), there is still relatively little research on its impact on aquatic and estuarine systems, compared to other pollutants and herbicides such as atrazine (*Ralph, 2000*; *McMahon et al., 2005*; *Diepens et al., 2017*) and heavy metals. Increasing agricultural application, together with glyphosate being licenced for the control of invasive aquatic weeds, has resulted in the detection of glyphosate in sediments and surface water across the globe, sometimes at concentrations above regulatory limits (*Peruzzo, Porta & Ronco, 2008*; *Matozzo, Fabrello & Marin, 2020*), although many uncertainties remain regarding its prevalence and persistence in aquatic systems. The exposure to and accumulation of glyphosate in estuarine and marine waters and sediments could as such significantly impact conservation efforts, including seagrass restoration if not measured and included in management actions (*Lewis & Richard, 2009*).

Comparatively little research has been done on the impact of this chemical on seagrasses, especially where long term exposure to sublethal levels is concerned. This is

concerning, as even low levels of herbicides and pollutants have been shown to sometimes have significant, even synergistic effects on plant growth and fitness (*Wilkinson et al., 2015*; *Diepens et al., 2017*; *Hughes et al., 2018*), which is likely to be compounded by the increased stressors plants will be exposed to under climate change (*Wilkinson et al., 2017*). Further research into this area is thus needed, with a specific focus on the potential impact of pollutants and herbicides on coastal ecosystems in general and seagrass in particular.

In this study, we examine the impact on *Z. capensis* morphology, photosynthetic pigment concentration and growth, of prolonged exposure to sublethal levels of a commercial glyphosate formulation under a range of glyphosate concentrations. We used glyphosate levels that are based on levels detected by environmental sampling of surface waters in watersheds where glyphosate is extensively used for agriculture, as well as on published and proposed legislated regulatory limits in surface waters. We predicted that even at the lowest levels of glyphosate exposure (250 µg/L) which falls within legally permitted levels in many countries, seagrass morphology and biomass will be negatively impacted, with potential effects on long term survival and fitness. These results will add to the body of work demonstrating the need to revise and standardise regulatory glyphosate limits (*Arcuri & Hendlin, 2019*). Our work also highlights the importance of accounting for glyphosate for seagrass conservation, particularly for restoration efforts where ecologically intact environments increase restoration success. For example, exposure to environmental glyphosate is likely an important consideration when evaluating population declines (*Diepens et al., 2017*), prior to initiating restoration efforts and when prioritizing sites for restoration and protection (*Devault, 2013*).

# MATERIALS AND METHODS

## Collection and acclimation

*Zostera capensis* ramets were collected in the Olifants river estuary (31.6982990S 18.2024690E; Fig. 1), from a 50 m transect along the intertidal zone. Shoots were collected at 5 m intervals, rinsed of sediment and stored in estuarine water in plastic resealable bags. Shoots were transported to the laboratory in an insulated box and planted within eight hours of collection. All collections were authorised under Department of Forestry, Fisheries and the Environment permit RES2021/68 and Cape Nature permit CN 35-87-15072.

Additional sediment was collected from an unvegetated area between seagrass beds and was well mixed by hand before being evenly distributed into plant containers. A total of 100 mL of beach sand was added to the top of the substrate to prevent disturbance and reduce turbidity when handling plants under experimental conditions. Containers were planted with one shoot from every transect point, resulting in 10 shoots per container. Shoots were selected to contain a set of roots and one leaf sheath with leaves. Shoot leaves were trimmed to 5 cm in length, and 1 cm of rhizome was planted, to approximate an equal starting biomass in every pot.

Polypropylene treatment tanks were prepared in advance, each containing 14 L artificial seawater (Red Sea salt™) at 31 ppt salinity, and two plant containers. The treatment tanks, each planted with 20 seagrass shoots, were placed in two, connected water baths to ensure

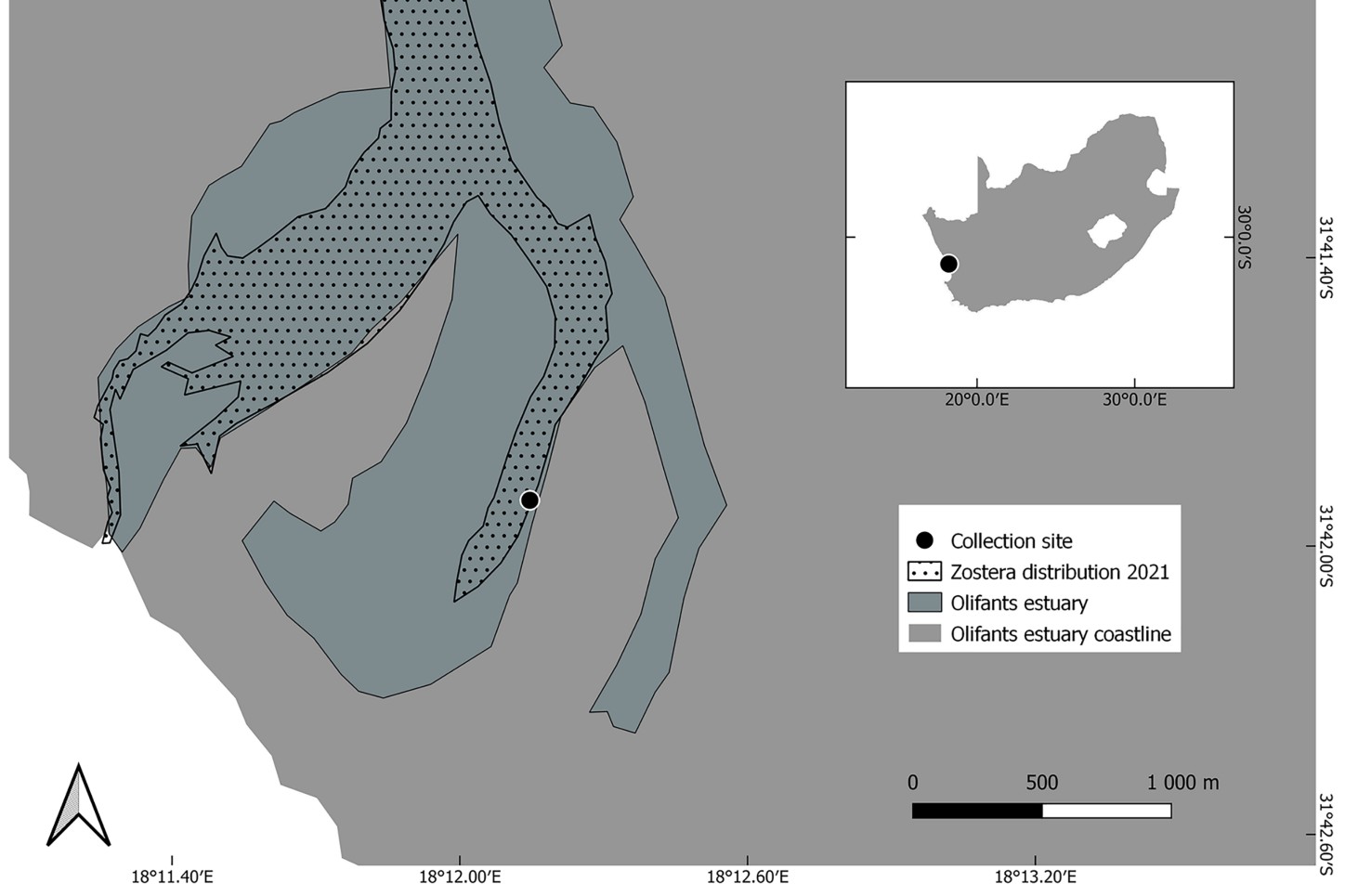

**Figure 1** *Zostera capensis* collection site, in the Western Cape province of South Africa, where the latter is shown as an inset.

consistent temperatures for all treatments. Polypropylene tanks were used rather than glass aquaria as glyphosate can be adsorbed onto glass surfaces, thus potentially decreasing concentrations of glyphosate (*Goscinny et al., 2012*).

Plants were acclimatised for four weeks at a water temperature of 21 °C (with daily measurements taken using a Thermochron iButton (iButtonLink, Whitewater, WI, USA)), while light was provided by a light array containing two T8 54 W actinic, and two T8 54 W 10,000 K daylight Odyssea fluorescent tubes, providing a surface irradiance of ~120 mmol/ m$^2$/s. A 16 h light/8 h dark cycle was maintained to provide >5 mol/m$^2$/day PAR, in line with the minimum light requirement in a morphologically similar *Zostera* species occupying a similar environmental niche, *Zostera muelleri* (*Bulmer, Kelly & Jeffs, 2016*). A 20% water change was performed per week and aeration and filtration provided by one F-800 Dolphin aquarium filter per tank. Salinity was measured twice a week using a handheld Red Sea™ Seawater refractometer and maintained between 31 and 33 ppt by adding reverse osmosis water if needed. During acclimation, a marked increase in the number of shoots, roots and rhizomes occurred and plants grew well, which was important
as glyphosate predominantly targets the actively growing parts of a plant (*Kanissery et al., 2019*). By the time glyphosate was added, all trimmed leaves had already senesced and the experiment was performed on new intact growth only.

After the acclimation period, treatment tanks were assigned to either control, 250, 750 or 2,200 μg/L treatment with Roundup™ using a randomised block design. The tested glyphosate concentrations were selected based on levels that have repeatedly been detected in environmental sampling (250 μg/L (*Nielsen & Dahllöf, 2008*; *Battaglin et al., 2014*)), legislated surface water limits (750 and 250 μg/L (*Mensah, Palmer & Muller, 2013*)) and levels experimentally shown to cause acute toxic effects in seagrasses (2,200 μg/L (*de Castro et al., 2015*)). Further, for studies comparing multiple treatments to a common control, it is recommended that the control group comprise at least 33% of the total samples (*Bate & Karp, 2014*), leading to a control group size of six tanks, compared to four tanks per treatment group.

Glyphosate based herbicides contain proprietary additives that may have significant additional toxic effects (*Gandhi et al., 2021*), although the exact nature of these additives is usually a trade secret. As these are the formulations used in practice, recent studies have aimed to use commercial formulations for toxicity testing, rather than pure glyphosate (*Mesnage et al., 2015*; *Defarge et al., 2016*; *Parlapiano et al., 2021*), to provide more representative effects such herbicides may be having in the environment. Roundup™ is the most commonly commercially available and applied (J. M. Dabrowski, 2019, personal communication) herbicide in South Africa and as such formed the basis of this study. A standard commercial agricultural product, Roundup Power 360 SL™ (glyphosate 360 g/L), was added to each individual tank at the beginning of the light cycle, at a dose calculated to achieve the desired concentration of the active ingredient, as glyphosate has been shown to have the greatest efficacy at this part of the circadian cycle in several plant species (*Belbin et al., 2019*).

A 20% water change was performed after seven days and again at 14 days to prevent fouling of seagrass by epiphytes. The experiment was conducted for 21 days, with chlorophyll, biomass and leaf morphology samples collected at the end of the experiment. Additionally, water samples were collected 24 h after treatment, to determine whether intended glyphosate concentrations had been achieved, and frozen at −20 °C until further analysis.

## Morphometric analysis

In order to determine the impact of glyphosate on *Z. capensis*, the following measurements were taken; leaf length and width, above and below ground biomass and shoot density, as these have been found to adequately describe leaf morphology in seagrasses (*Kuo & Hartog, 2006*). Ten shoots were randomly selected from each of the two planted pots contained in every treatment tank (pot A and pot B) and the three longest leaves/shoot were measured for length and width, using Vernier callipers and leaf area calculated. Pot A samples were measured and harvested on day 21 and pot B samples on day 23, due to unforeseen circumstances. This was captured in the 'Sample' variable, which showed no statistically significant difference between groups on analysis. Plant material was washed

and separated into above and below ground biomass, dried for eight hrs at 60 °C; dry matter weight was determined using an Explorer Ohaus electronic scale. Shoot density, above and below ground dry matter biomass, leaf length, width and area were compared using analysis of variance.

## Glyphosate analysis

Water samples, collected 24 h after initiating treatment, were immediately frozen at −20 °C and maintained at that temperature until further analysis. Samples were analysed for glyphosate and its breakdown product, aminomethylphosphonic acid (AMPA), using the method described by *Wang et al. (2016a)* for seawater. Briefly, this entails derivatization of the sample with 9-fluorenylmethylchloroformate (FMOC-Cl), separation with high performance liquid chromatography (HPLC) and detection of glyphosate and AMPA by their fluorescence. At the Central Analytical Facility at Stellenbosch University, a Waters ACQUITY Ultra Performance Liquid Chromatography (UPLC) was coupled to a Xevo Fluorescence Detector (FLR) (Waters, Milford, MA, USA) and used for UPLC-FLR analysis. Glyphosate and AMPA were separated using a Waters ACQUITY BEH C18, 100 × 2.1 mm, 1.7 μm particle size column at 60 °C and a flow rate of 0.4 mL/min. An injection volume of 2 μl was used and the mobile phases consisted of 5 mmol/L ammonium acetate in water (Solvent A) and Methanol (Solvent B). The following gradient was used: 90% A, 0–0.54 min; 90–30% A, 0.5–6.0 min; 30–1% A, 6.0–7.0 min; 1–90% A, 7.0–8.0 min; 90% A, 8.0–10.0 min. The FLR was set at 265 nm (excitation) and 315 nm (emission).

Blank samples were run concurrently using distilled water adjusted to a salinity of 31 ppt. Calibration curves were created using an artificial seawater matrix, as *Wang et al. (2016a)* found no significant difference between natural seawater and artificial seawater calibration curves. The limit of detection (LOD) was determined to be 0.5 μg/L and limit of quantification (LOQ) 1 μg/L.

## Chlorophyll and carotenoid content analysis

Leaf samples were collected at harvest and frozen at −20 °C until analysis. Chlorophyll and carotenoid content was extracted using a modified method, with 100% acetone used instead of 70% acetone (*Pocock, Król & Huner, 2004*). A total of 0.05 g of wet weight leaf sample (three samples from each tank) was ground in 1 mL of chilled acetone. The extract was centrifuged at 3,100 g for 5 min at a temperature of 10 °C. The supernatant was collected and chilled for 3 h at 4 °C, after which 50 μL of extract was added to 950 μL of 100% acetone. This was centrifuged at 2,000 g for seven minutes at a temperature of 4 °C. The absorbance of the supernatant was measured using a Biotec Power-wave HT. Total chlorophyll a and b, carotenoids and total chlorophyll concentration were calculated following *Porra (2002)*, with details on equations provided in the Supplemental Table B.1.

## Statistical analysis

Although Roundup^TM was added to treatment tanks at levels calculated to achieve 250, 750 and 2,200 μg/L of the active ingredient glyphosate, there was a large variation in the concentration of glyphosate measurable in the tanks after 24 h, even within treatment

groups, likely due to differences in adsorption to sediment and/or absorption into the plants themselves between different tanks. A total of 24 h post exposure, the 250 µg/L samples measured 106–181 µg/L (mean 148 µg/L), the 750 µg/L samples measured 370–750 µg/L (mean 533 µg/L) and the 2,200 µg/L samples measured 770–1,800 µg/L (mean 1,333 µg/L) (see Supplemental Material Table A.1).

In addition, the time weighted average exposure over the three-week treatment period is lower than initial exposure levels (see Supplementary Material Tables A.2 and A.3). Nonetheless, initial nominal values were used for all analyses, as glyphosate is known to need only 6 h of exposure for toxic effects to become apparent, with toxic effects taking up to three weeks to manifest according to the technical datasheet (*Henderson et al., 2010*). In addition to glyphosate, AMPA was detected during this experiment, with maximum levels of 25 µg/L at 24 h.

Analyses of variance (ANOVA) were performed on the different treatment groups, investigating effects on leaf area, leaf length, leaf width as morphological measurements, as well as above and below ground biomass as measures of growth. Groups were clustered according to the initial nominal dose of Roundup™, as well as on glyphosate concentration measured 24 h after administration. For leaf length, area and leaf width, 360 samples were measured for the control group and 240 for each of the three treatment groups.

While leaf width followed a normal distribution, leaf length, leaf area, chlorophyll and biomass data followed a gamma distribution. Leaf area and leaf length data were square root transformed to approximate a normal distribution of the residuals, for use in the ANOVA. Photosynthetic pigment data (control group $n = 54$, each treatment group $n = 36$) and biomass data (control group $n = 12$, each treatment group $n = 8$) could not be transformed and were analysed using a Kruskall-Wallis test, while residuals for leaf width did not need transformation.

All statistical analyses were carried out in R studio (*RStudio Team, 2020*), using the "car" and "ggplot2" libraries. The Aov() and Anova() functions were used, together with a Tukey Kramer post-hoc test for data following a normal distribution, and a Kruskall-Wallis and Dunn test performed on data that could not be transformed. Data were tested for normality using a Cullen and Frey graph from the R package "fitdistrplus", and equal variance confirmed by the Levene test in "car" package. Tank 12 appeared as an outlier when inspecting the morphology data and two data sets were analysed; one including tank 12 and one without. Analysis on the reduced dataset is shown in the text (where n for the 750 µg/L treatment group was reduced to 180), while analysis of the full dataset is available in the Supplemental Materials. Biomass and chlorophyll data were analysed on the full dataset.

Data which had to be transformed for analysis are represented in their original form as boxplots in the Supplemental Material, while data with a normal distribution are represented with a plot of mean and confidence intervals in the text.
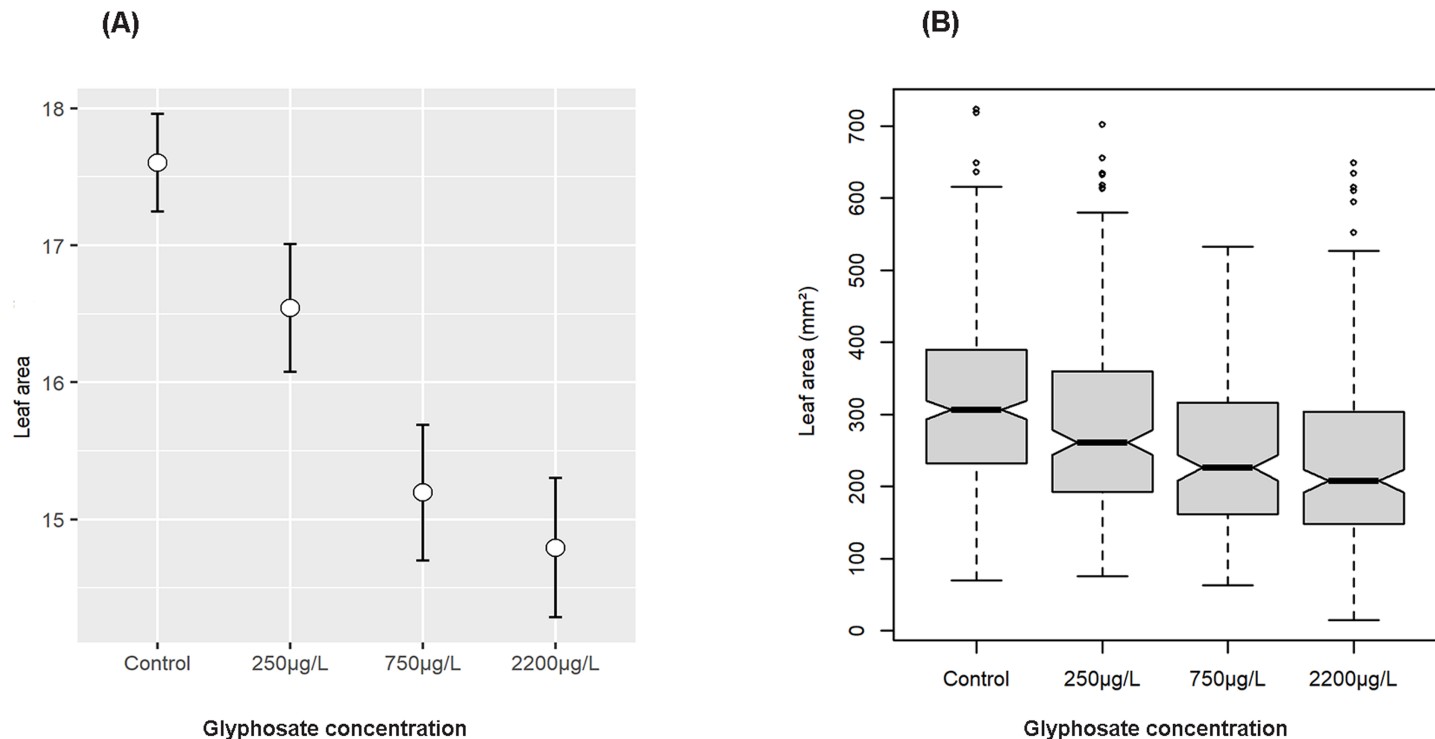

**Figure 2** Effect of nominal glyphosate concentration on mean and confidence intervals of transformed leaf area (A) and notched boxplot showing median and interquartile range of leaf area (B) showing the decrease in leaf length with increasing glyphosate concentration. Median leaf area decreased by 11% when treated with 250 μg/L of glyphosate and by 27% when treated with 2,200 μg/L.

## RESULTS

### Morphometric analysis

Analyses of leaf area, length and width performed on the reduced dataset, grouped according to nominal administered glyphosate levels, are presented. Glyphosate concentration had a strong, statistically significant negative effect on leaf area (F value 37.014, $p = 2.2e{-}16$, Figs. 2A and 2B), although an interaction between water baths and glyphosate treatment was also detected (F value 10.625, $p = 7.002e{-}07$; Type III ANOVA).

The same was true for leaf length (F value 33.030, $p = 2.2e{-}16$, Figs. 3A and 3B), which also had an interaction with water bath (F value =12.104, $p = 8.696e{-}08$; Type III ANOVA). When the 24 h measured glyphosate concentrations instead of nominal administered treatment concentrations were used in analysis, this interaction fell away, indicating that the interaction may have been due to differing initial adsorption or absorption of glyphosate in the two baths, despite all efforts to standardise conditions.

Glyphosate concentration had a strong, statistically significant negative effect on leaf width (F value 31.225, $p = 2.2e{-}16$, Fig. 4), although an interaction between the two water baths and the glyphosate treatment was also detected (F value 8.135, $p = 2.35e{-}05$) (Type III ANOVA). Analyses on the full morphology dataset are provided in the Supplemental Material, showing the same trend as the curated dataset but with a weaker (although still statistically significant) effect on the 750 μg/L treatment group.

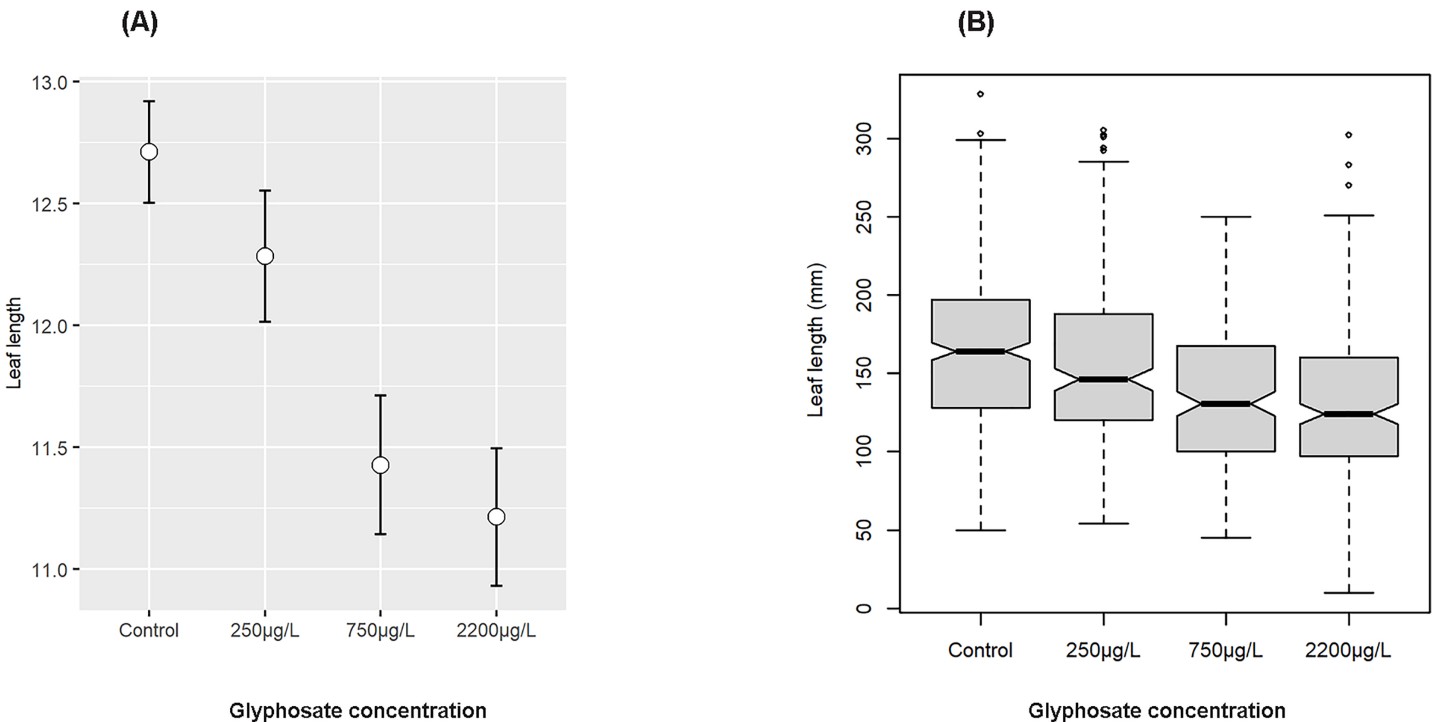

**Figure 3** Effect of nominal glyphosate concentration on mean and confidence intervals of transformed leaf length (A) and notched boxplot of median and interquartile range of leaf length (B) showing the decrease in leaf length with increasing glyphosate concentration.

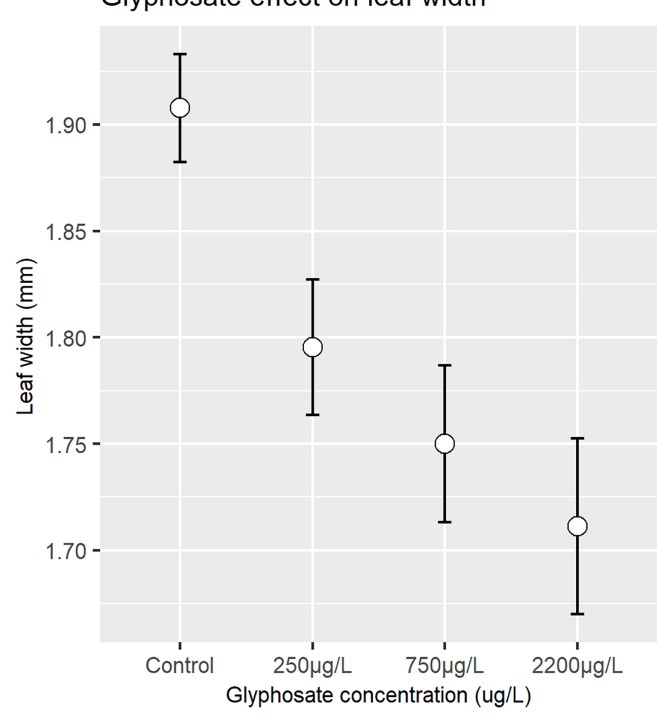

**Figure 4** Mean and confidence intervals of leaf width (mm) as affected by nominal glyphosate concentration.

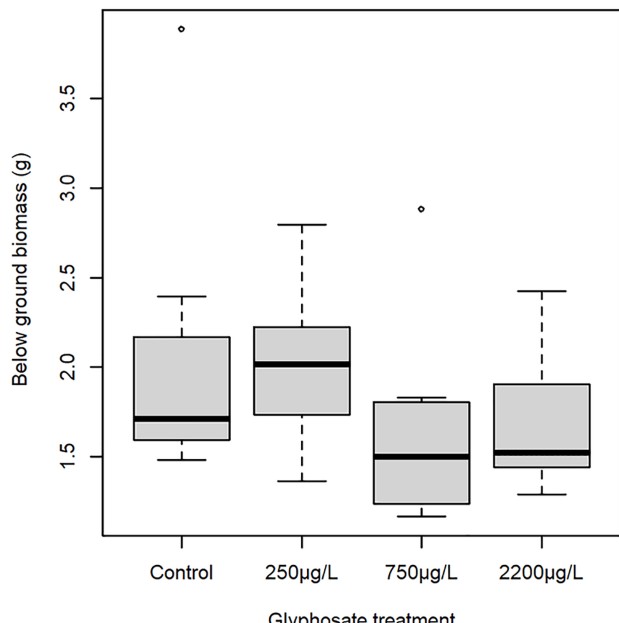

**Figure 5 Boxplot showing effect of nominal glyphosate concentration on (A) above and (B) below ground biomass.** Median above ground biomass declined with increasing glyphosate concentration, showing an 11% decrease when treated with 250 μg/L of glyphosate and 31% decrease when treated with 2,200 μg/L, while below ground biomass showed no trend.

Shoot density between treatment and control groups was not statistically significant, and no trend was detected in the data (data not shown).

Above and below ground biomass data did not pass Q-Q plots for normality, therefore a Kruskal-Wallis rank sum test was performed. For above ground biomass, the effect was significant at the 90% confidence level (chi-squared 7.219, df = 3, *p*-value = 0.0653). A Dunn test was subsequently performed, only returning a significant p value for the difference between the control and 2,200 μg/L groups. However, a definite downward trend was observed in in the data (Fig. 5A), and the lack of statistical significance may be due to the smaller sample size for biomass data. The effect for 24 h glyphosate concentration was similar (data not shown).

Below ground biomass showed no pattern (Fig. 5B) and there was no statistically significant difference between different treatment groups (Kruskal-Wallis test, chi-squared 5.397, df = 3, *p*-value = 0.145). All data are available at https://github.com/vonderHeydenLab/van-Wyk-et-al_seagrass-and-glyhosate_raw-data.

## Chlorophyll analysis

There were no statistically significant differences between groups for total chlorophyll content, chlorophyll a or chlorophyll b (Supplemental Material Table B.2). Carotenoid content was negligible.

## DISCUSSION

### Brief overview of biodiversity impacts and global glyphosate legislation

Glyphosate is a post-emergent, non-selective herbicide that targets the shikimate pathway by inhibiting the enzyme 5-enolpyruvylshikimate-3-phosphate synthase (EPSPS) (*Steinrücken & Amrhein, 1980*). This enzyme is found in plants, algae, some bacteria, and fungi, but is absent in mammals. In plants, glyphosate disrupts folate, aromatic amino acid and lignin synthesis–leading to disrupted protein production, chlorophyll degradation and reduced growth—with maximum effect in areas of rapid metabolism such as root tips and shoot apices (*Gomes et al., 2014*). It is also a strong metal chelator, forming complexes with minerals such as iron and magnesium, leaving them unavailable to the plant (*Cakmak et al., 2009*). In seagrasses this is of particular concern, as iron availability reduces sulphide stress and promotes seagrass growth in anaerobic environments, such as that found under *Zostera* meadows (*Holmer, Duarte & Marbá, 2005*). In addition, there is evidence that glyphosate exposure can lead to increased susceptibility to disease, especially fungal root rot, in plants (*Duke et al., 2012*; *van Bruggen et al., 2021*).

Although glyphosate based herbicides were initially widely marketed as a safe alternative to older herbicides, there is increasing concern about their wider ecological impact (*Mesnage et al., 2015*; *Sikorski et al., 2019*; *Matozzo, Fabrello & Marin, 2020*; *Corrales, Meerhoff & Antoniades, 2021*) as well as their impact on human and animal health ( *Peillex & Pelletier, 2020*; *van Bruggen et al., 2021*; *Vandenberg et al., 2017*), with research revealing potential impacts on reproduction, renal health, carcinogenesis and even transgenerational effects due to epigenetic changes (*Milesi et al., 2021*). The potential long term effects on ecosystems may be severe, especially when considering the possibility of synergistic effects with other chemicals also present in the environment (*Gamain et al., 2018*).

This is of special relevance for organisms, such as aquatic plants, occupying lower trophic levels, where chronic sublethal effects on reproduction and fitness have the potential to disrupt entire ecosystems (*Githaiga et al., 2019*). In this regard, the limited research available on the impact of glyphosate and its formulations on marine and coastal systems is concerning, with indications that estuarine and marine organisms may be particularly sensitive to its effects. For example, it has recently been shown that glyphosate levels as low as 1 µg/L can lead to total loss of recruitment in the canopy forming marine macroalga, *Carpodesmia crinita* (*de Caralt et al., 2020*), with potential long term implications for population persistence.

In seawater, glyphosate has been shown to persist for between 47 and 315 days under low light conditions (*Mercurio et al., 2014*), while breakdown in sediment may take considerably longer than in water (*Wang et al., 2016b*) and sediment levels are often orders of magnitude higher than water levels in catchment areas where glyphosate is frequently applied, leading to organisms potentially being exposed to this herbicide for extended periods of time. Despite these concerns (*Arcuri & Hendlin, 2019*), regulatory limits vary immensely between countries and have been adjusted upwards in several jurisdictions as

**Table 1 Glyphosate levels detected in recent environmental surface water samples.**

| Country | Environment | Glyphosate concentration | Reference |
|---|---|---|---|
| USA | Groundwater | 2 µg/L | (*Battaglin et al., 2014*) |
| | Precipitation | 2.5 µg/L | |
| | Large rivers | 3 µg/L | |
| | Streams | 73 µg/L | |
| | Wetlands | 302 µg/L | |
| | Ditches | 427 µg/L | |
| | Sediment | 476 µg/kg | |
| Argentina | River | 125 µg/L | (*Bonansea et al., 2017*) |
| Argentina | River | Up to 0.7 µg/L | (*Ronco et al., 2016*) |
| | Sediment | Up to 3,000 µg/kg | |
| Argentina | River | Up to 1.80 µg/L | (*Primost et al., 2017*) |
| | Sediment | Up to 3,294 µg/kg | |
| Germany | Baltic Estuaries | 0.028 to 1.690 µg/l | (*Skeff, Neumann & Schulz-Bull, 2015*) |
| | Sediment | 2 to 12 µg/kg | |
| Denmark | Stream | 287 µg/L | (*Nielsen & Dahllöf, 2008*) |
| South Africa | Stream | 0.4 µg/L | (*Horn, Pieters & Bøhn, 2019*) |

the demands of agriculture have increased. In the USA, where agriculture is dependent on massive herbicide use, the maximum residue level of glyphosate in drinking water is set at 700 µg/L (*U.S. Environmental Protection Agency, 1995*). Guidelines for the protection of the environment are set in a similar range. For example, the Australian water quality guideline for the protection of 99% of fresh water species sets a limit of 370 µg/L glyphosate in surface waters (*Mercurio et al., 2014*), while in South Africa, *Mensah, Palmer & Muller (2013)* suggested short and long term freshwater glyphosate limits of 250 µg/L and 2 µg/L respectively for the protection of indigenous aquatic biota. However, to date, no guidelines for glyphosate levels in South African waters have been legislated and it is rarely tested for.

## Glyphosate impacts on an estuarine ecosystem engineer

There has been little research into the presence of agricultural pesticides (herbicides, insecticides and fungicides) in African waters (*Porter et al., 2018*; *Horn, Pieters & Bøhn, 2019*; *Curchod et al., 2020*), although there is a substantial body of research documenting the presence of herbicides and other agricultural chemicals in surface waters in other parts of the world (*Wilkinson et al., 2015*; *Diepens et al., 2017*) (Table 1). In this study, the glyphosate levels tested were based on environmentally realistic levels, as have been detected repeatedly in environmental sampling globally (Table 1), and also on the values proposed by regulatory authorities as allowable (*U.S. Environmental Protection Agency, 1995*; *Mensah, Palmer & Muller, 2013*). Ours is the first African study on the potential impacts of glyphosate on aquatic vegetation and suggests that even at levels within the suggested South African regulatory limit of 250 µg/L, and significantly below the USA regulatory limit of 700 µg/L, glyphosate was found to have a detrimental effect on seagrass leaf area, length, and width. The effect was strongly significant and increased with

increasing levels of glyphosate, with declines in median leaf area of ~11% at the lowest level of glyphosate exposure and ~27% at the highest level, with median leaf length and width showing similar trajectories (Figs. 2–4). As expected, a decrease in above ground biomass was also noted. Further, the time weighted glyphosate exposure levels in this experiment were less than half the initial nominal dose of glyphosate administered (see Supplemental Material), which may indicate that *Z. capensis* is sensitive to even lower levels of glyphosate exposure than indicated by the initial doses. As our study only investigates the effect of glyphosate on a subsample of a single population of *Zostera capensis*, future studies should include specimens from multiple populations to determine if the effect holds true for genetically distinct populations (that have been demonstrated in *Z. capensis*; *Phair et al., 2019*), especially those from less pristine environments which may have acclimatised to herbicide exposure. In future, more studies should be carried out to identify clear chronic effect concentrations (EC10 values) for seagrass growth, based on measured concentrations.

In our study, *Z. capensis* leaves became consistently shorter and thinner as glyphosate concentration increased, with a concomitant decrease in above ground biomass. This is in contrast to research by *Nielsen & Dahllöf (2008)* which found an increase in above ground biomass and no effect on leaf morphology after three days of exposure to glyphosate levels of 169 and 1,600 µg/L. However, the lack of effect seen in *Nielsen & Dahllöf (2008)* may be due the short time frame used. Most research on glyphosate effects have been carried out over short time frames (<7 days), while glyphosate may take up to three weeks to have its full effect, even at commercial doses of 3.6 g/L (*Kanissery et al., 2019*). As such, the cumulative effect of long-term exposure to environmentally realistic levels are likely very different and future glyphosate impact studies will benefit from longer, more realistic exposure experiments.

Our findings of negative impacts on leaf morphology in aquatic plant species are not unique, as *Sikorski et al. (2019)* recorded similar levels of decline in response to similar concentrations of glyphosate-based formulations in the duckweed, *Lemna minor*. However, even with no significant differences in shoot density or chlorophyll concentration between treatment and control groups in *Z. capensis*, it is likely that the morphological changes observed may lead to decreased photosynthetic area available to the plant, in turn leading to decreased carbohydrate storage. This is supported by research showing that reducing above ground biomass during the growing season leads to lower carbohydrate storage in *Zostera noltii*, while exposure to low levels of glyphosate is also known to disrupt carbon metabolism in plants (*Mateos-Naranjo & Perez-Martin, 2013*). As lower carbohydrate stores are associated with decreased winter survival in seagrasses (*Govers et al., 2015*) and as decreased energy reserves may lead to lower resilience to stressors such as higher turbidity, freshwater run-off and disease, this may result in decreased fitness and survival over time. Reduced above ground biomass may also result in lower total carbon sequestration (*Soissons et al., 2018*) which may be of concern, as seagrass is estimated to store between 4.2 and 19.9 petagrams (Pg) of carbon globally (*Fourqurean et al., 2012*), and blue carbon is increasingly being investigated as a tool for climate mitigation (*de Castro et al., 2015*), including in South Africa (*Adams et al., 2019*).

Although there was no statistically significant difference for below ground biomass in this study, it may that longer time frames would be needed to observe changes in this variable. The effect of chronic low grade glyphosate exposure on carbohydrate accumulation in rhizomes also invites further investigation, as chronic exposure to other herbicides like diuron has been shown to negatively affect below-ground starch reserves, even at concentrations lower than those that affected above ground leaf growth (*Negri et al., 2015*).

## Future considerations for assessing herbicide impacts on the conservation of seagrasses

Our results support the assertion that in many countries, regulatory limits on the level of glyphosate-based herbicides allowed in surface waters may be set too high. In the view of increasing evidence of environmental harm at even relatively low levels, regulatory limits need to be reviewed and urgent attention given to alternative methods of weed control. Within the context of seagrass conservation, our work suggests that measuring glyphosate (and likely other herbicides) in for example, areas earmarked for restoration, is an important contributor to increasing restoration success and increasing both seagrass density and cover.

In addition, there is evidence in plants that reproductive life stages are more affected by glyphosate exposure than mature and seedling stages (*Boutin et al., 2014*), leading to reduced reproduction in plants exposed during the reproductive stage. As glyphosate is accumulating in sediments worldwide, associated with increasing application of the herbicide together with its slow breakdown in sediments, the effect of environmentally realistic levels in sediment on seed production (*Boutin et al., 2014*), germination and biomass of new shoots should be investigated. Although it has been shown that faba beans, oats and turnip rape had delayed germination and decreased shoot biomass after being grown in soil with a glyphosate concentration of 4.8 mg/kg (*Helander et al., 2019*), the effects in seagrasses and other marine macrophytes are still unknown. Given that many seagrass restoration projects rely on repopulating meadows by planting seeds (*Unsworth et al., 2019*; *Govers et al., 2022*), and that high levels of glyphosate in sediments may contribute to poor germination outcomes in certain plant species, determining sediment glyphosate concentrations may well be vital to restoration and conservation success.

In addition, synergistic effects of other environmental stressors and herbicide exposure should be further investigated. For example, research suggests that glyphosate may become less effective in some plant species at higher temperatures and $CO_2$ concentrations (*Matzrafi et al., 2019*); however there remain limited insights and this has not been tested for seagrass species. Further, evidence suggests that sublethal combinations of chemicals, which on their own would have no effect, may have synergistic effects in combination (*Nielsen & Dahllöf, 2008*), potentially leading to reduced fitness, increased susceptibility to stressors such as increased water temperature and disease and even increased mortality (*Gamain et al., 2018*; *Hughes et al., 2018*). For example, *Nielsen & Dahllöf (2008)* showed that glyphosate mixed with other herbicides, at low levels such as those detected in the environment, act synergistically, having a far greater effect on growth and chlorophyll concentration of *Zostera marina* than would be suggested by individual toxicity testing.

However, this is the only study we could find examining the effects of glyphosate in combination with other pollutants on seagrasses. Due to the increased chemical load in the environment, the impact of low-level glyphosate exposure when combined with environmentally realistic levels of other pollutants therefore warrant urgent further investigation in aquatic systems globally.

In conclusion, additional studies are urgently needed to investigate the effects of chronic long-term exposure to environmentally realistic levels of herbicides, including glyphosate, on seagrasses, particularly looking at effects on survival, carbohydrate storage and reproduction. Of special concern is the impact of cumulative stressors, such as increases in water temperatures and turbidity expected with climate change, as well as the potentially synergistic effects of multiple pollutants on seagrass resilience and survival in a rapidly changing world. This study improves our understanding of the interaction of anthropogenic pressures such as pollution on coastal aquatic ecosystems and the importance of accounting for these in conservation efforts of seagrasses worldwide.

## ACKNOWLEDGEMENTS

The authors wish to thank C. Hui for their advice on the statistical analyses and A. Ndhlovu for support during field lab work.

### Funding

This project was funded by the National Research Foundation South Africa through Marine and Coastal Research Grant Number 116048. The DSI/NRF Research Chair in Shallow Water Ecosystems (UID 84375) supported J. Adams. The funders had no role in study design, data collection and analysis, decision to publish, or preparation of the manuscript.

### Grant Disclosures

The following grant information was disclosed by the authors:
National Research Foundation South Africa through Marine and Coastal Research: 116048.
The DSI/NRF Research Chair in Shallow Water Ecosystems: UID 84375.

### Competing Interests

The authors declare that they have no competing interests.

### Author Contributions

- Johanna W. van Wyk conceived and designed the experiments, performed the experiments, analyzed the data, prepared figures and/or tables, authored or reviewed drafts of the article, and approved the final draft.
- Janine B. Adams conceived and designed the experiments, authored or reviewed drafts of the article, and approved the final draft.

- Sophie von der Heyden conceived and designed the experiments, authored or reviewed drafts of the article, and approved the final draft.

## Field Study Permissions

The following information was supplied relating to field study approvals (*i.e.*, approving body and any reference numbers):

All collections were authorised under Department of Forestry, Fiserhies and the Environment permit RES2021/68 and Cape Nature permit CN 35-87-15072.

## Data Availability

The data is available at GitHub: www.github.com/vonderheydenlab/van-Wyk-et-al_seagrass-and-glyhosate_raw-data; Van Wyk, JW, Von der Heyden, S, & Adams, J. (2022). Conservation implications of herbicides on seagrasses: sublethal glyphosate exposure decreases fitness in the endangered *Zostera capensis* [Data set]. Zenodo. https://doi.org/10.5281/zenodo.7156564.

## Supplemental Information

Supplemental information for this article can be found online at http://dx.doi.org/10.7717/peerj.14295#supplemental-information.

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
