# Peer review of "Conservation implications of herbicides on seagrasses: sublethal glyphosate exposure decreases fitness in the endangered Zostera capensis"

_PeerJ, doi:10.7717/peerj.14295_

## Round 0.1 · original submission · Major Revisions

Dear Authors,

The two independent reviewers have just finished revising your manuscript.

Based on their comments, there are some crucial flaws that must be addressed appropriately before the manuscript may be considered for publication in PeerJ. Therefore, I suggest the authors reply to these reviewers with a point-to-point rebuttal letter and highlight all changes in the text in the revised manuscript.

We thank the authors for considering the Peer J for submission.

Best Regards
Carmen Arena

Reviewer 1 ·

Basic reporting

The article must be written in clear and technical language. In the presented manuscript generally happens, despite this I suggest some changes:
• For the corresponding author Add an address under the name
• Line 53: Remove ‘Concerningly’.
• Line 54: REPLACE ‘meadows are in decline globally, with Waycott et al. (2009) documenting a decline in cover’ WITH ‘meadows are globally in decline, as demonstrated by Waycott et al. (2009) who documenting a decline in cover’.
• Line 61: ‘ Although eutrophication and decreased light availability’. ADD A REFERENCE for light availability or re-modulate the concept if is included in previews references ( i.e. Burkholder et al., 2007; Waycott et al., 2009; Mvungi and Pillay, 2019).
• Line 65-66: REPLACE ‘African species and conditions are poorly represented in the global literature on the effects of pollutants on seagrasses’ WITH ‘ the effects of pollutants on seagrasses African species are poorly represented in the global literature ’.
• Line 73: REPLACE ‘distribution’ with a more comprehensive word, if conceivable.
• Line 83: ‘ to other pollutants and herbicides such as atrazine and heavy metals’. ADD A REFERENCE.
• Line 97: ADD A REFERENCE at the end.
• Line 122-123: REPLACE ‘(-31,6982990 18,2024690)’ WITH ‘31,6982990 S 18,2024690 E).
• Line 138: I suggest adding a figure to better explain the experimental design.
• Line 148: ADD A SENTENCE that describes the check of the salinity during the experiment, If occurred.
• Line 165-172: MOVE in the introduction.
• Line 248: ADD A REFERENCE to the sentence.
• Line 332: REPLACE ‘seagrass’ WITH ‘seagrasses’.
• Line 350: ‘disrupt entire ecosystems’. ADD A REFERENCE.
• Line 363: REPLACE ‘of’ and ‘has’ WITH ‘for’ and ‘have’.
• Line 366: ‘Environmental Protection Agency’. ADD A REFERENCE.
• Line 370. In the cited paper, Mensah et al calculated a species sensitivity distribution (SSD) of 250 mg/L for short-term glyphosate-based herbicides.
• Line 408: REMOVE final reference.
• Line 425: REPLACE ‘parameter’ WITH ‘variable’.
• Acknowledgments: do not Acknowledge founders in this section.
• Line 469-477: Following the ‘instructions for authors’ it is better to include conclusions in a separate paragraph.
• Supplementary material Figure A.1: REPLACE ‘significanly’ WITH ‘significantly’.
• Supplementary material Figure A.2: REPLACE ‘significanly’ WITH ‘significantly’.

Experimental design

The experimental design is quite clear and linear. However, I ask the corresponding author to better explain the two points described below
1) Line 133: ‘Shoot leaves were trimmed to 5 cm in length’. It is possible to assume that trimming does not cause any additive effects, other than those of glyphosate, on leaves? Please add reasons and, possibly, references.
2) Line 157: Reference studies Nielsen and Dahllöf, 2008; Battaglin et al., 2014) cited in the manuscript were conducted in Denmark and USA, respectively. Furthermore, Table 1 indicates, for South Africa, the paper of Horn et al., (2019) as a reference to glyphosate levels in the area. In this article, they show results of glyphosate concentrations in two rivers sited near two farms. They detected a maximum value of 0.4 µg/L for the glyphosate concentration. Finally, Mensah et al. calculated a species sensitivity distribution (SSD) of 250 mg/L for short-term Roundup exposure, to provide suggestions for South African water quality guidelines (SAWQGs).
Please explain better, in discussions, the choice of the 250 µg/L value as the minimum exposure value.

Validity of the findings

I thank you for providing the row data. I suggest checking the 'Glyphosate experiment Biomass May 2021’ sheet, it seems that the glyphosate concentrations at 24h have been rounded unlike what is reported in the table A.1 of supplementary material

Additional comments

The manuscript is written in understandable and professional English. However, I suggest a general overhaul of the language and I have noticed some changes that could be made. Furthermore, it is necessary to add more references in some parts of the article, to better define the concepts of the sentences. Finally, a better explanation of some details of the experimental design(i.e. leaves trimming, and exposure concentrations choice) and a row data check, are needed.

Reviewer 2 ·

Basic reporting

Dear authors,
Thank you for the opportunity to review “Conservation implications of herbicides on seagrasses: sublethal glyphosate exposure decreases fitness in the endangered, Zostera capensis”.
The article is generally clear and concise, using unambiguous and technically correct text. The structure is conform to PeerJ standard format and all the materials (e.g. figures and tables) are relevant to the content of the article.
Despite this, the introduction has many details that are unnecessary and sometimes challenging to follow (lines 81-89). On the other hand, the last paragraph on the study aims (lines 107-117), could be better articulated by providing more explanations about why their work is needed in a conservation and restoration framework: e.g., for revising and standardizing legislated regulatory glyphosate limits, improving the protection of the seagrass in the study region, prioritizing sites for restoration. This could be done broadening the last sentence of the introduction (lines 115-117).

Experimental design

Methods are well described and investigation is conducted rigorously, allowing to assess sublethal thresholds of glyphosate for the seagrass Z. capensis. The study design provided relevant insights to improve the understanding of the interaction between human pressures and aquatic ecosystems.
Nonetheless, I have a question regarding the stage of life under which the experiment was conducted. Since the authors discuss the increased effects of glyphosate during the reproductive life stage, I would ask if they considered the possibility to include in their experiment plants in that phase too. This would have resulted in a more complete view of the impact of this herbicide on the study species, strengthening the findings achieved with this work.

Validity of the findings

Conclusions are appropriately stated and supported by results, even if I could not find figures 2b and 3b among the provided materials. Apart from this, I have two points of concern regarding the robustness of the analysis performed.
The first one is about lack in site replication. I believe that the obtained results are supported by a weak sample design which, taking into account only one site and one transect, can be insufficient to describe the variability in the response of Z. capensis to glyphosate. I would suggest considering this weakness in the discussion, at least.
The second one is about the statistical analysis performed on the morphological response variables. I wonder why the authors opted for three different ANOVA when testing for differences among treatments, instead of performing a single MANOVA/PERMANOVA analysis, taking into account leaf area, length and width at the same time.

Additional comments

I commend the authors for their very clear analysis on the effects of glyphosate on the seagrass Z. capensis. The manuscript is written in a professional and unambiguous language and all the material which accompanies the text is relevant, supporting in a very transparent way the analysis performed. I believe that the findings achieved by the authors are crucial to stepping forward legislation on glyphosate limits worldwide. Also, they considered a seagrass species still scarcely investigated, advancing knowledge for its conservation and restoration in a data-poor region. The most concerning point in my view is the robustness of the authors’ findings since I believe that site replication is strongly needed to validate the manuscript results.
Minor comments:
I would delete the comma from the title
Lines 83-89: sentences challenging to follow, I would suggest rephrasing them.
Line 112: I would specify what “lower” is referred to.
Line 117: I would suggest expanding this concept and supporting this consideration with some literature.
Lines 151-152 / 175-176: I think that these sentences require some references.
Line 308: the word “in” is repeated twice.
Lines 448-450: hard to follow, I would suggest rephrasing the sentence.
Discussion should start with what results have shown and continue with general considerations on glyphosate impacts. For this reason, I would suggest inverting the order of the first two paragraphs of the discussion section.

---

## Round 0.2 · accepted · Accept

Dear Authors,

Thank you for your submission to PeerJ.

I am writing to inform you that your manuscript - Conservation implications of herbicides on seagrasses: sublethal glyphosate exposure decreases fitness in the endangered Zostera capensis - has been Accepted for publication.

Congratulations!

Reviewer 1 ·

Basic reporting

I thank the authors for almost totally accepting my suggestions. I believe that the changes to the original text are satisfactory.

Experimental design

I thank the authors for having answered my questions. I believe that the arguments were well proposed and describes in a good way the potential effects of the trimming and the exposure concentrations choice.

Validity of the findings

I thank you the authors for the check.

Additional comments

I thank the authors for improving the text in the requested manner.

Reviewer 2 ·

Basic reporting

Dear Prof. Arena,

I’ve carefully read the authors’ response and believe that the manuscript has much improved in its current version. Also, the authors addressed all my concerns and suggestions. For this reason, I recommend this article for publication.

Yours sincerely

Experimental design

No additional comments

Validity of the findings

No additional comments

Additional comments

No additional comments